

# Optimized MobileNetV3: a deep learning-based Parkinson's disease classification using fused images

Sukanya Pechetti and Battula Srinivasa Rao

School of Computer Science and Engineering, VIT-AP University, Andhra Pradesh, India

## ABSTRACT

**Background and Objective:** Parkinson's disease (PD) is a progressive neurological condition that manifests motor and non-motor symptoms. Early in the course of the disease, PD patients frequently experience vocal difficulties. In the beginning, preprocessing procedures were used with multi-focus image fusion to enhance the quality of input images. It is essential to diagnose and treat PD early to ensure that patients live healthy and productive lives.

**Methods:** Tremors, rigidity in the muscles, slow movement, difficulty balance, and other psychological symptoms are some of the disease's symptoms. One of the critical mechanisms supporting PD identification and assessment is the dynamics of handwritten records. Several machine-learning techniques have been researched for the early detection of this disease. Yet the main problem with most of these manual feature extraction methods is their poor performance and accuracy.

**Results:** This cannot be acceptable when discovering such a chronic condition. For this purpose, a powerful deep learning model is suggested to help with the early diagnosis of Parkinson's disease. Therefore, we proposed MobileNetV3-based classification. To enhance the classification performances even more, the MobileNetV3-based approach was optimized by the Improved Dwarf Mongoose Optimization algorithm (IDMO).

**Conclusion:** The Pyramid channel-based feature attention network (PCFAN) chooses the critical features. The efficiency of the approaches is tested using the PPMI and NTUA datasets. Our proposed approach obtains 99.34% accuracy, 98.53% sensitivity, 97.78% specificity, and 99.12% F-score compared to previous methods.

# INTRODUCTION

Parkinson's disease (PD) is a nervous system diagnosis that gradually impairs brain function. It affects the brain's dopamine-producing cells to perish (*Rezaee et al., 2022*). Consequently, it has an impact on the body's motor system. Pesticide exposure and a history of head injuries are the leading causes of PD. Research indicates that people with PD are more prone to smoking, drinking alcohol, having high cholesterol, or consuming excessive food (*Gunduz, 2021*; *Masud et al., 2021*). The signs of PD include delayed movement, tremors, imbalance, poor posture, and speech distortion. Dopamine-boosting

Corresponding author
Battula Srinivasa Rao,
sreenivasvitap@gmail.com

medications aid in the recovery of PD patients. Early diagnosis helps in starting the treatment at the initial stage and lessens the intensity of the patient's symptoms. Most methods for diagnosing the illness involve analyzing the patients' gait signs and speech (*Polat & Nour, 2020*). Moreover, sensors placed beneath their feet record the patients' forces to keep track of changes in body posture (*Balaji, Brindha & Balakrishnan, 2020*). One of the contemporary systems for grading and categorizing the various stages of PD is the UPDRS. Tremors can impact a person's movement, arms, head, and even gait and are frequently associated with PD's core symptoms (*Sivaranjini & Sujatha, 2020*; *Solana-Lavalle & Rosas-Romero, 2021*; *Amato et al., 2021*). If the disease is not diagnosed at an early stage, the patients are at risk of losing life (*Lee et al., 2021*). Professionals, however, face difficulty detecting the condition early as it does not show up on examinations and exhibits symptoms similar to those of many other illnesses at this time. Both industry and academia have been working on developing sophisticated computational tools that can diagnose the disease automatically (*Yang et al., 2021*; *Moon et al., 2020*; *Balaji et al., 2021b*).

Computer-aided diagnosis (CAD) technologies are widely used in research to analyze various diseases (*Kamble, Shrivastava & Jain, 2021*). Machine learning (ML) based methodologies in such situations have recently shown exceptional results in various medical applications, including diagnosing and treating diseases like Alzheimer's and breast, brain, and esophageal cancer (*Sharma, Singh & Kaur, 2021*). These investigations (*Williams et al., 2020*) extracted clinically significant features from speech signals using various speech signal processing methods, which were subsequently fed into several artificial learning systems to get accurate PD classification results. While the most popular methods in PD classification include random forest (RF), artificial neural network (ANN), support vector machine (SVM), and k-nearest neighbors (KNN), they are also helpful because of their clarity and simplicity (*Al-Sarem et al., 2021*; *Priya et al., 2021*; *Senturk, 2020*). The qualities of the data features gathered affect how well the algorithms work. While it can be challenging to locate sufficient features individually to capture the essential aspects of speech (audio) data, the fundamental properties of the data can be manually recognized using a deep learning technique (*Balaji et al., 2021a*; *Xu & Pan, 2020*).

This research provides a distinctive deep learning-based classification method to address the abovementioned challenges. Multi-fusion can be included in the preprocessing phase to enhance the effectiveness. The PD classes are categorized with the help of optimized MobileNetV3.Our key contributions include creating an effective deep-learning architecture for accurate identification, thoroughly assessing it, and comparing it to the state-of-the-art, demonstrating consistent improvements in various performance metrics. The remaining sections are structured as follows: The related research that has been conducted in recent years on Parkinson's detection is highlighted in "Related Works". The materials and approaches used for the presented work are described in "Proposed Methodology". The description of the dataset utilized for this investigation is presented in "Result and Discussions", followed by an explanation of the suggested framework and a comparison with currently available models. The proposed conclusion is summarized in "Conclusion" in the final paragraph.

## Novel contribution

The significant key contributions of this article are as follows,

- In the pre-processing stage, we first utilized the image contrast enhancement algorithm to enhance the difference of a given input image. Next, the mean filter was used to eliminate the noise from the input image. Then, we blended the contrast-enhanced image and filtered the image into a single image as a fused image employing the multi-scale morphological gradient method.
- For extracting the features of the preprocessed image, we presented a new Pyramid channel-based feature attention network (PCFAN) that employed a multi-stage design with attention blocks at every step.
- To classify the levels of PD, such as mild, moderate, and severe, we employed the MobileNetV3 technique with the Dwarf Mongoose Optimization algorithm to improve the classification accuracy.
- Several ablation experiments have been performed on the PPMI and NTUA datasets. The experimental results showed that the suggested network outperformed the state efficiency compared to all other methods.

## RELATED WORKS

This section summarizes current studies on the machine learning algorithms used in PD categorization and provides information on the most recent deep learning techniques.

To assess gait data and develop a DNN classifier for Parkinson's disease, *El Maachi, Bilodeau & Bouachir (2020)* introduced a 1D convolutional neural network (1D-Convnet). The suggested framework processes 18 parallel 1D data from foot sensors measuring the vertical ground response force (VGRF). A total of 18 parallel 1D-Convnets that correspond to system inputs make up the first segment of the network. In the second component, which is a fully linked network, the outcomes of the 1D-Convnets are concatenated to create the final categorization. Our tests showed that the suggested strategy highly detects PD from gait data. The proposed technique had a 98.7% accuracy rate. Using deep learning, *Loh et al. (2021)* developed a 2D-CNN model to diagnose Parkinson's disease automatically. The computerized detection of PD algorithm used in this study classified spectral images into unaffected PD patients with or without dopamine-producing medications and healthy controls using the suggested 2D-CNN model. The suggested model obtained a high accuracy of 99.46% for multi-categorization employing tenfold cross-validation.

*Ortiz et al. (2019)* suggested a technique for PD identification utilizing isosurfaces-based characteristics and CNN. They employed a convolutional neural network (CNN) model based on LeNet-5 and AlexNet to identify isosurfaces and extract descriptive information. Similar to how contour lines connect places of equal height, isosurfaces link voxels with the specified intensity or value. The effort culminated in creating a categorization system that used supervised learning *via* CNN architectures to categorize DaTSCAN pictures. *Karaman et al. (2021)* created deep CNNs to automatically identify PD using voice signals

obtained from biomarkers. Data pre-processing and fine-tuning-based transfer learning processes comprised two primary stages of the established CNN algorithms. They investigated whether combining a speech dataset from a large dataset with transfer learning model fine-tuning techniques could improve PD identification. The findings showed that the suggested deep CNN model, which combines transfer learning with a fine-tuning strategy, can diagnose PD with a 91.17% accuracy.

Using one-dimensional convolutions and bidirectional gated recurrent units (BiGRU), *Diaz et al. (2021)* developed a unique categorization framework to investigate the utility of handwriting's sequential information for detecting Parkinson's symptoms. In this research, the raw sequences and derived features were subjected to one-dimensional convolutions; the generated sequences were then fed to BiGRU layers to produce the final categorization. The recommended approach outperformed other existing alternatives when compared. *Olivares et al. (2020)* developed an improved ELM by utilizing the Bat Algorithm that increases the machine learning method's training phase to enhance the accuracy while lowering or maintaining loss in the learning phase. With input weights and bias values, the approximation technique simultaneously defines an optimal vector. It is intended to optimize the ELM's training phase to produce the best categorization model. When compared with existing approaches, the presented approach yields greater performances.

Employing deep learning (DL) approaches, *Vyas et al. (2022)* offered two unique methods. Convolution neural networks (CNNs) in 2D and 3D that were trained on axial-plane MRI data were utilized. Four pre-processing stages, N4 bias correction, histogram matching, z-score normalization, and picture scaling, were performed to increase the algorithm's effectiveness. Using the information, the 3D model could categorize the test data with 88.9% accuracy and 0.86 area under the curve (AUC).

### Limitations

However, this study may have some limitations:

- Due to the computational difficulty of two-dimensional CNN models, training is time-consuming.
- Substantial amounts of memory in the computer are necessary as the method may crash If the memory requirement for training a method outweighs the number of shots.
- The proposed model's generalizability could be constrained by the limited sample size of the PD dataset employed in this investigation.

To overcome these problems, a Pyramid channel-based feature attention network (PCFAN) is used to extract the relevant features to reduce computational complexity. In our proposed method, we used a large number of images for training purposes for better prediction with sufficient memory. While comparing with our proposed approach, the existing approaches yield less prediction accuracy and high detection rates. Also, it takes high computational time. A 99.34% accuracy rate is achieved by this system, offering results that are equivalent to those of the recently suggested approach. As a result, it can be

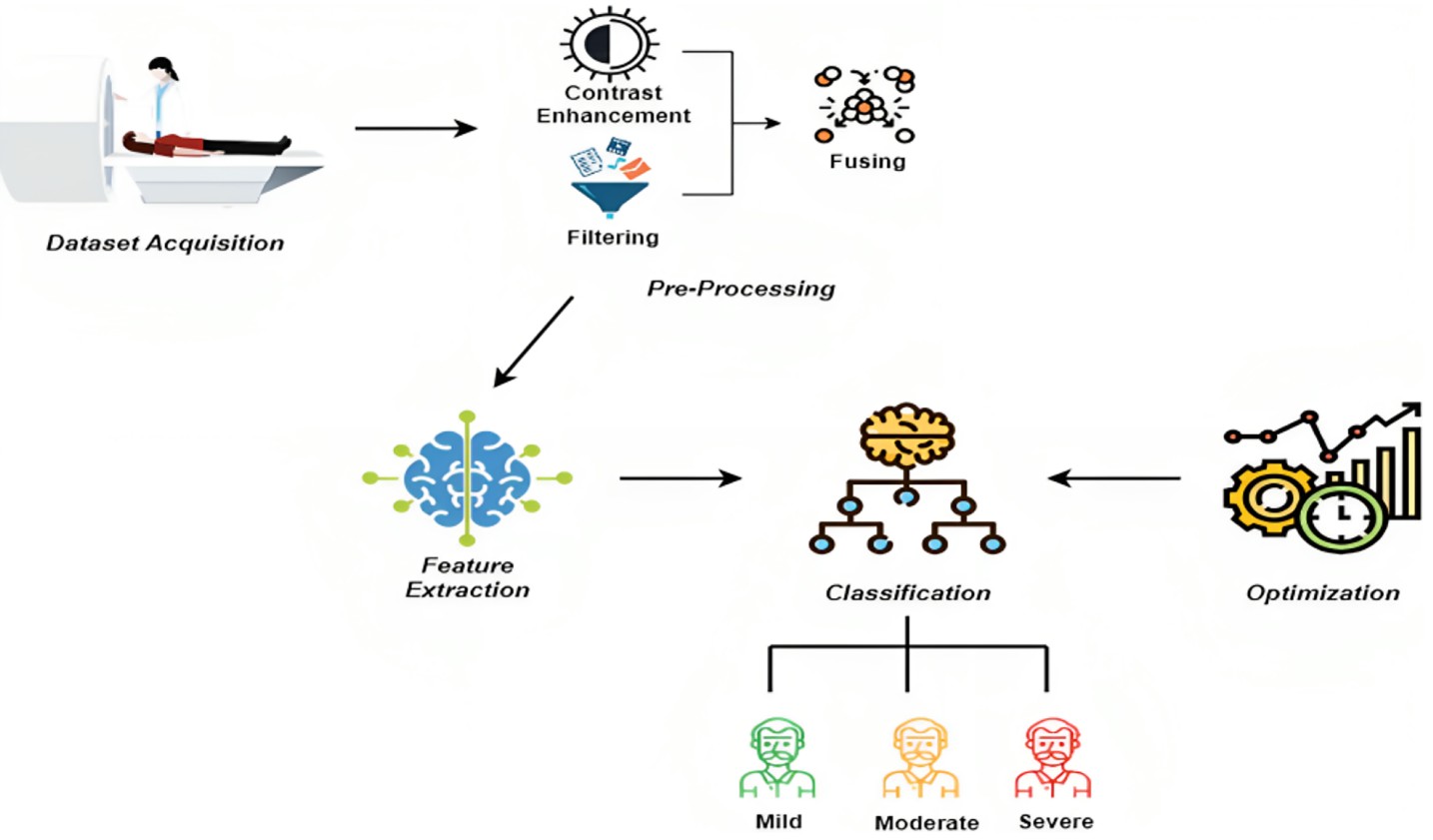

**Figure 1 Proposed architecture diagram.**

concluded that the input data is significantly reduced when models are computed while maintaining the relevant data, producing excellent classification accuracy with a low computational effort.

## PROPOSED METHODOLOGY

The proposed methodology comprises three phases: preprocessing, feature extraction, and classification. In the beginning, during the pre-processing step, we first used the image contrast enhancement algorithm to improve the contrast of the provided input image, and then a mean filter was used to remove the noise.

After employing a multi-scale morphological gradient approach, the contrast-enhanced image and the filtered image were combined into a single image to create the fused image. We introduced a novel Pyramid channel-based feature attention network (PCFAN) that used a multi-stage design with attention blocks at each level to extract the features from the preprocessed picture. Finally, the images were classified as mild, moderate, and severe with the help of MobileNetV3. The IDMO algorithm optimized the classification approach hyperparameters. Figure 1 represents the architecture diagram of the proposed methodology.

## Image preprocessing

This subsection provides a detailed explanation of the proposed preprocessing algorithm. Some contrast abnormalities have been removed using preprocessing methods to improve classification results. We have created two copies of the input images to accomplish this. Contrast adjustment was done to enhance the visualization of the needed region in the initial copy. The image's contrast was altered using the highest and minimum values of the image pixels.

## Image contrast enhancement algorithm

A novel approach has been suggested by *Ying et al. (2017)* to give accurate contrast enhancement, and it was utilized to build the enhancement dataset. The weight matrix for picture fusion was first built using lighting prediction methods. The algorithm then functioned as follows. The camera response model was made available, allowing for the combining of numerous exposure photos. The optimal exposure ratio was then discovered for a decent exposure of the synthetic image in areas where the source image was underexposed. Further, a weight matrix was used to combine the input image and the created image to create a superior image. Equations (1) and (2) provide the essential formulas that the algorithm used. The images were integrated as in Eq. (4) to create an image with all pixels.

$$R^c = \sum_{i=1}^{N} W_i P_i^c \tag{1}$$

while N indicates the quality of images, $P_i$ the i-th image of the exposure set $W_i$ represents the weight map of the image, c for the three-color channel index, and R for the enhancement's outcome. $P_i$ is calculated from Eq. (2).

$$P_i = g(P, k_i) \tag{2}$$

g is referred to as the brightness transform function (BTF), and $k_i$ is the exposure ratio. In our investigation, the BTF was the beta-gamma connectivity method from Eq. (3).

$$g(P, K) = \beta P^\gamma = e^{b(1-k^a)} P^{(k^a)} \tag{3}$$

The variables a, b, and k of the camera could be used to compute the parameters β and γ. As in the initial investigation, we used a constant parameter (a = 0.3293; b = 1.158). In the conclusion of the method, Eq. (4) was used to produce the enhanced image.

$$R^c = WP^c + (1 - W)g(P^c, k) \tag{4}$$

## Mean filtering

Using numerous picture flattening patterns for graphic convolution processing is typical for picture de-noising in the spatial domain. The foundation of mean filtering is replacing a pixel's single grey value with the total of all the surrounding pixels' grey values. According

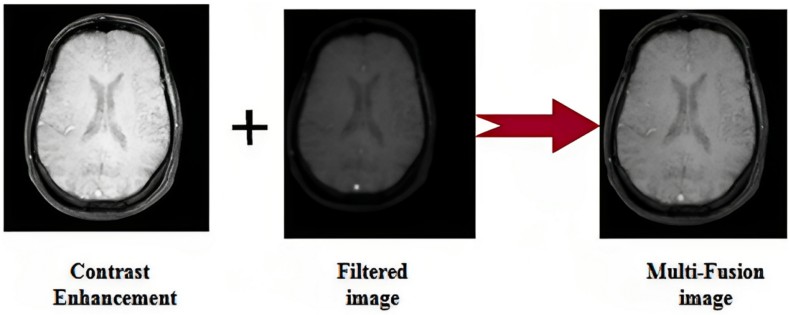

**Figure 2 Sample image.** Photo credit: B. Srinivasa Rao.

to the following process, an image is created for a pixel point (a, b) in a source image with f (a, b), while its surrounding area S contains M pixels:

$$G(x, y) = \frac{1}{M} \sum_{(i,j \in S)} f(a, b)(a, b) \notin S \tag{5}$$

## Multi-scale morphological gradient

The efficient operator, the multi-scale morphological gradient (MSMG), extracts gradient information from a picture to show the contrast level in a pixel's nearby areas. As a result of this, the MSMG technique is quite effective and used for edge detection and picture segmentation. MSMG has been employed in multi-focus image fusion as a focus measure. The following details are used for MSMG. A multi-scale structural element is described as

$$SE_j = SE_1 \oplus SE_2 \oplus \dots \oplus SE_N, j \in \{1, 2, \dots, N\} \tag{6}$$

SE1 is a notation for a fundamental structural component. By using the morphological gradient operators from picture f, the gradient feature $G_t$ can be expressed.

$$G_t(x, y) = f(x, y) \oplus SE_t - f(x, y).SE_t \tag{7}$$

while the morphological operators for dilation and erosion are denoted by $\oplus$ and (.) accordingly. Here, t refers to the number of scales. By estimating the weighted total of gradients across all scales, one may derive the multi-scale structuring element from the gradient feature. The same preprocessed images are shown in Fig. 2.

$$M(x, y) = \sum_{t=1}^{N} w_t.G_t(x, y) \tag{8}$$

Here, the weight of the gradient in the t-th scale is denoted by

$$w_t = \frac{1}{2t + 1} \tag{9}$$

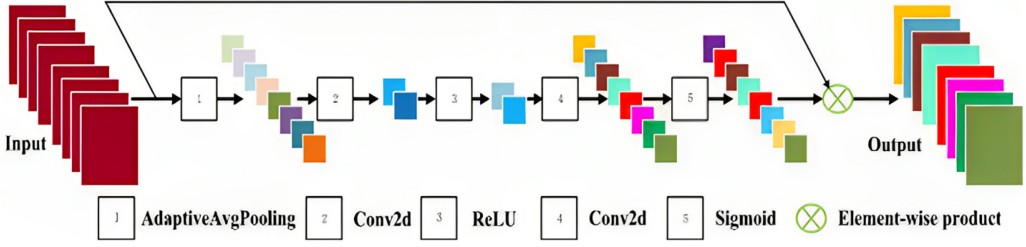

**Figure 3** Framework of Pyramid channel-based feature attention network (PCFAN).

## Pyramid channel-based feature attention network (PCFAN)

### FrameWork

This study presents a PCFAN model for feature extraction by fusing channel attention and pyramid operation advantages. Three modules comprise this PCFAN: an image reconstruction module, a module for PCFA, and a module for extracting features from three scales. Three phases comprise the three-scale feature module: Two Resblocks plus a $3 \times 3$ convolution layer comprise the initial feature extraction stage. At this stage, feature maps include 32 more channels (or depth).

They increase the depth of the feature maps to 64 and 128 and reduce the resolution of the feature maps by half, respectively. Unlike earlier studies, which mainly used the third stage's output features, the PCFA, made of several channel-attention blocks, provides data from all three stages' output features. It is possible to extract spatial and channel-dimension information using the channel-attention block. Then, a network for image reconstruction with just one CNN layer is used to recover the original clean image.

**Channel attention block:** To make sure that the network collects additional features, the channel attention strategy is employed in this study to look at the dependencies between feature channels.

The framework of PCFA is shown in Fig. 3. Concatenation operation is performed by fc at position $(i, j)$, $[v_1, v_2, ....v_c]$ and as a result, $\mu$ is the concatenation of $v_k (k = 1 ... C)$. These processes can record how the aggregated features depend on one another through channels. This is represented as:

$$\vec{f} = \sigma(\phi_2(\eta(\phi_1(\mu)))) \tag{10}$$

The convolution layer, sigmoid activation function, and ReLU are denoted by $\phi$, $\eta$, and $\sigma$. The goal $\phi_1$ is to decrease the input feature channels. The features are first activated by ReLU$\eta$, then with a convolution layer 2, they are expanded to their original width. The final output feature $F_{out}$ is attained by

$$F_{out} = \vec{f} \otimes f \tag{11}$$

while $\otimes$ is an element-wise product, and the original feature is represented as $f$.

**PCFAN:** This model can retrieve features from various CNN layers and merge those characteristics simultaneously to produce more valuable features. These techniques, however, typically employ an intuitive fusion process, such as addition or concatenation.

In order to integrate the advantages of the feature pyramid and the channel attention mechanism, we presented the PCFA approach.

PCFA comprises two layers of upsampling, two levels of concatenation, and four channel-attention blocks. In PCFA, top-down and bottom-up pathways are both available. The suitable channel attention blocks for the bottom-up path-way are filled with features from three tiers. PCFA reconstructs layers with higher spatial resolution from the down-top pathway using their semantically richer layers. Combining information from the top-down and bottom-up routes makes it possible to express traits effectively and comprehend their relative importance at different levels. The complementary information among low-level and high-level attributes can, therefore, be fully utilized by PCFA for extraction.

### Loss function

The suggested network is optimized using the Mean Square Error (MSE) loss $\mathcal{L}_{mse}$ and the Edge loss $\mathcal{L}_{edge}$, two loss functions. The MSE loss is employed to assess differences between image outcomes regarding pixel-wise aspects. The MSE is explained as follows:

$$L_{mse} = \frac{1}{CWH} \sum_{c=1}^{c} \sum_{i=1}^{w} \sum_{j=1}^{H} \left( I_{c,i,j}^{clear} - \bar{I}_{c,i,j}^{extracted} \right)^2 \tag{12}$$

### Classification

In order to classify Parkinson's disease after successfully identifying brain images in scans, we suggested utilizing MobileNetv3 deep learning. The CNN family, known as MobileNet, was created by a team of Google, Inc. researchers for the purpose of classifying images. MobileNet included several innovative ideas through its numerous iterations to decrease the number of parameters while maintaining excellent classification accuracy. Compared to several other CNN architectures of equivalent size, MobileNet performs well in terms of accuracy because of the MADDS (multiply-add operations). MobileNetV3 (*Zivkovic et al., 2022*), in particular, has the top-1 accuracy among the other models. Investigating the proposed approach for this categorization challenge was primarily motivated by this.

MobileNet is made of bneck blocks, a collection of construction blocks. Figure 4A shows the overall MobileNet architecture, whereas Fig. 4B shows the intricacies of a bneck block. MobileNetV1 used depth-wise convolutional operations in place of typical convolutional methods. In Fig. 4B, a residual link between the input and results tensors is shown. Then, as depicted in Fig. 4, the creators of MobileNetV3 added both the compression and expansion stages to the start and finish of every bneck block. This setup is called an "Inverted Residual Block" (IRB), as the residual connections only make limited output and input tensor connections.

The IRB idea contributed to further lowering the model's computational expenses. In order to further reduce computations (such as ReLU), the authors used linear activations rather than non-linear activation functions after filtering the input and output tensors. The SE module was included by the authors to finalize the MobileNetV3 concept.

The SE module also features a function of the h-swish added by the authors. The description of the Swish activation function is as follows:

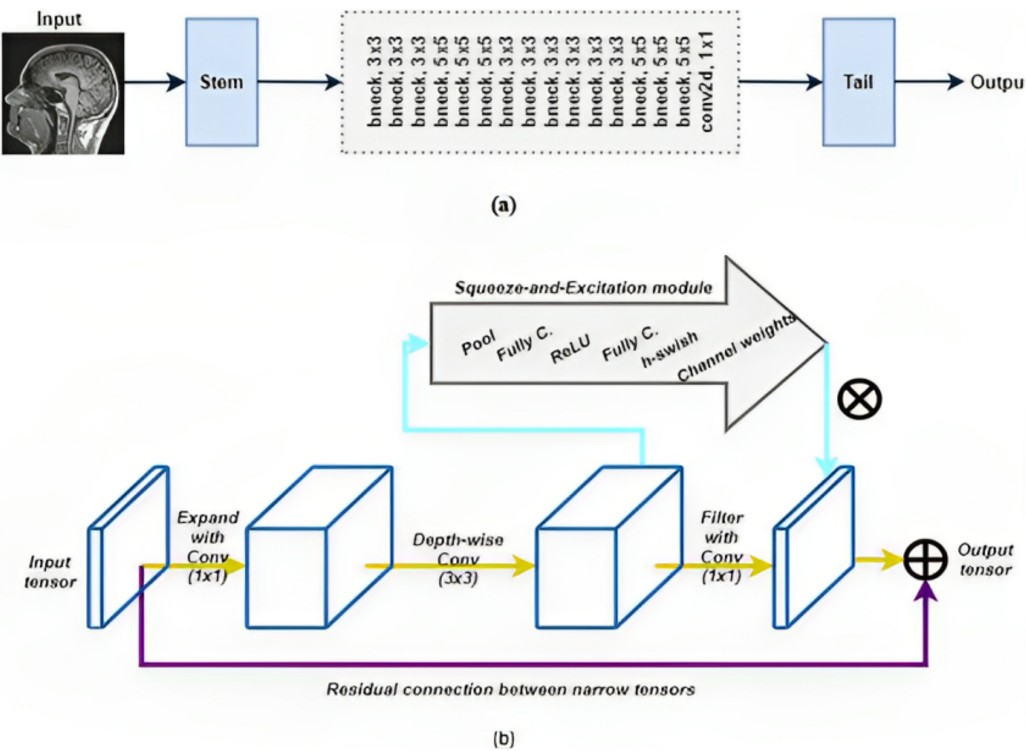

**Figure 4 A collection of bneck blocks are combined to create MobileNet functions.** (A) A high-level summary. (B) bneck block illustration. Photo credit: B. Srinivasa Rao.

$$h - swish(x) = x \frac{ReLU6(x + 3)}{6} \tag{13}$$

The description states that a bneck block constructs a feature map optimized using SE components and connection residuals. Our decision to use the bneck block as the fundamental model of an architecture resembling the UNet was motivated by this.

## Improved dwarf mongoose optimization algorithm (IDMO)

This phase provides details on the IDMO algorithm working process.

### The IDMO mode

The Improved Dwarf Mongoose Optimization algorithm (IDMO) is suggested to improve DMO exploration and exploitation. The DMO is modified in three straightforward yet efficient ways by this optimization technique. Alphas are chosen by the IDMO rather than the DMO, which chooses them solely based on their computational load. The IDMO's alpha selects the mongoose based on physical fitness, and a new operator is presented to control the alpha's mobility. This increases the IDMO's exploration and exploitability.

Secondly, randomization changes the scout group motions to diversify the search and investigate previously unexplored places. The suggested approach obtains optimization in three stages they are babysitters, forage area, and abundant food source. The search agents

are the individual mongooses, described as an n-by-d matrix. The modified alpha guides the group to unexplored territory during the exploration phase by following the modeled processes.

Randomization is used to change the scout group motions in order to diversify the search process and explore previously unexplored locations. Once the requirement for babysitter swap is satisfied and babysitters are switched, the exploitation is accomplished. At this stage, the obtained solution is upgraded in order to yield superior results.

## Population initialization

According to Eq. (14), a matrix of potential dwarf mongooses (X) is used to stochastically initialize the IDMO population. Among the optimization issue's upper bound (U) and lower bound (L), the population vector lies.

$$X = \begin{bmatrix} x_{1,1} & x_{1,2} & x_{1,d-1} & x_{1,d} \\ x_{2,1} & x_{2,2} & x_{2,d-1} & x_{2,d} \\ & : & x_{i,j} & : \\ x_{n,1} & x_{n,2} & x_{n,d-1} & x_{n,d} \end{bmatrix} \tag{14}$$

while n represents the entire amount of dwarf mongooses in a mound, Every $X_{i,j}$ in Eq. (15) reflects the location of the jth population's dimension.

$$X_{i,j} = rand \times (U - L) + L \tag{15}$$

## Alpha group

Babysitters are subtracted from the overall amount of dwarf mongooses to get the total population dimensions for this group. The dwarf mongoose that is the fittest is chosen to be the alpha, as shown by the alpha symbol in Eq. (16).

$$\alpha = \min(fit_1, fit_2 \ldots \ldots, fit_n) \tag{16}$$

Peep's vocalizations are used by the female alphas to keep them integrated. As specified in Eq. (16), the IDMO moves about the problem space while searching. It initially identifies as the strongest dwarf mongoose in the family and leads the other members of the pack towards a prospective food source. In contrast to the DMO, which relies solely on the vocalization of the alpha to sway the behavior of the other dwarf mongoose, this situation deviates from it. The IDMO's exploration and exploitability are improved by operator *a*. The IDMO uses the location of the alpha to establish the location of the other mongoose.

$$X_{a+1} = \alpha + phi * rand * (Xa - X_u) \tag{17}$$

$$\omega = e^{-4*(C_{iter}/Max_{iter})^2} \tag{18}$$

## Scout group

Scouts are responsible for selecting an appropriate sleeping mound as dwarf mongooses are semi-nomadic and are reported to never return to their sleeping mounds. Due to the dwarf mongooses' propensity for congregating around plentiful food sources, the scouts'

fitness level is considered when choosing a prospective sleeping mound. Consequently, the fittest scout is chosen. As stated in Eq. (19), the scouts are simulated.

$$X_{a+1} = \alpha + phi * rand * (X_u - X_{v)/2} \tag{19}$$

### The babysitters

Equation (20) provides the exchange standard for the babysitter. Once the requirement is met, the counter is reset to 0, and the swapped babysitters communicate with the dwarf mongooses. By doing this, they could create mongooses better suited to their environment rather than starting from scratch, as in DMO. By multiplying L by the current iteration and CF, L is reset if it reaches zero.

$$X_{a+1} = (Xb + rand * (\alpha - (Xu + Xv)/2 * br \tag{20}$$

while $CF = \left(1 - \dfrac{C_{iter}}{Max_{iter}}\right)^{\left(2\frac{C_{iter}}{Max_{iter}}\right)}$ directing the dwarf mongooses' overall volitional movement, $X_b$, $X_v$, and $X_u$ are chosen at random to take the position of the babysitters, and br denotes the birthrate.

It makes the alpha selection overhead simpler, and the IDMO's computing complexity is significantly decreased. Dwarf mongooses forage under the leadership of the alpha female, which initiates the optimization process. The nest is tended by a small group known as the babysitters. Finding sufficient food sources resembles the IDMO's exploratory phase. The exploitation phase of the IDMO is represented by this hypothetical scenario. The search region is further investigated and exploited by looking for a sleeping mound at night.

## RESULT AND DISCUSSIONS

This section investigates the specifics of the analytical results obtained using the suggested methodology based on evaluation criteria like sensitivity, accuracy, specificity, and F1 score. The five steps in this research include dataset description, preprocessing, feature extraction, classification method, and analysis outcomes. This study categorized PD diseases using the PPMI and NTUA datasets. Two sets of data were created from the full dataset: a training set that contained 80% of the data and a validation set that contained 20% of the data. Several tests were conducted using various network configurations. Network configuration options were kernel size, batch size, stride, and padding. During each epoch of model training, validation, and training accuracy statistics were recorded. The model was tested on the test set after every training cycle. It was observed that the categorization accuracy varied across all experiments above 99%. The output samples of proposed methodology is shown in Fig. 5.

### Experimental setting

Windows 10 was used on an Intel i5 2.60 GHz processor with 16 GB of RAM. The investigations were conducted using Python, KERAS, and TensorFlow against the backdrop of the Anaconda3 environment. In this research, the PPMI and NTUA datasets

**Figure 5 Result of proposed methodology.** Photo credit: B. Srinivasa Rao.

were used for validation to calculate the performances of our suggested strategy. The data samples were divided into two groups, one served as the training dataset and was used to build a classifier. In the second step, the classifier was assessed using the testing dataset.

## Dataset description

The Parkinson's Progression Makers Initiative (PPMI) collection contained photographs of patients and controls accessed for the research. An international network of clinical sites was where the PPMI was conducted. One goal of the PPMI was to gather medical, biological, consumer, and imaging data in order to hasten the establishment of biomarkers of PD progression. The ultimate objective of these biomarkers was to be employed in therapeutic investigations. For this investigation, T1-weighted MR images from PPMI were chosen.

With the use of a 1.5–3 Tesla scanner, these images were produced. It takes about 20 to 30 min to complete the scan. Axial, sagittal, and coronal views were used to acquire the three-dimensional (3D) sequence of T1-weighted MR images with a slice thickness of

1.5 mm or less. Approximately 6,500 images in the dataset. From the dataset, we utilized 4,550 (70%) images for training purposes, and the remaining 1,950 (30%) images were utilized for testing purposes.

## NTUA

The NTUA Parkinson dataset contains MRI, DaT scans, and clinical information from PD-affected patients. There were about 42,000 photos that could be used for research purposes. The frames per sequence and resolution of the T1, T2, and Flair MRI image samples in this dataset varied for each image. From the dataset, we utilized 29,400 (70%) images for training purposes, and the remaining 12,600 (30%) images were utilized for testing purposes.

## Evaluation criteria

To evaluate the prediction capabilities of the classifiers, evaluation measures were required. The following metrics were employed to evaluate the proposed method.

**Accuracy:** Accuracy is the percentage of accurate forecasts among all made predictions.

$$Accuracy = \frac{TP + TN}{TP + FP + TN + FN} \tag{21}$$

**Sensitivity:** Sensitivity, sometimes called true positive rate (TPR) or Recall, evaluates a system's tendency to anticipate the future positively.

$$TPR = \frac{TP}{TP + FP} \tag{22}$$

**Specificity:** Specificity, commonly called true negative rate (TNR), evaluates a system's capacity to forecast negative outcomes correctly.

$$TNR = \frac{TN}{TN + FP} \tag{23}$$

**Precision:** Precision, called positive prediction value (PPV), assesses a system's capacity to generate only relevant outcomes.

$$\Pr ecision = \frac{TP}{TP + FP} \tag{24}$$

**F-Measure:** The harmonic mean of precision and recall was computed by F-measure.

$$F - Measure = 2 * \frac{\text{Re}call * Precision}{\text{Re}call + precision} \tag{25}$$

## Evaluation of training and testing time

The batch size, learning rate, momentum, and weight decay were each 32, 0.03, 0.9, and 0.01. A 0.01 learning rate was used initially. The learning rate reached saturation in the ReLu layer. The quantity of epochs was also a vital training parameter, as there was a

| Table 1 Optimized hyperparameter used for training. | |
|---|---|
| **Hyperparameter** | **Optimized value** |
| Optimizer | Adam |
| No. of epochs | 200 |
| Batch size | 32 |
| Momentum | 0.9 |
| Decay | 0 |
| Learning rate | 0.001 |

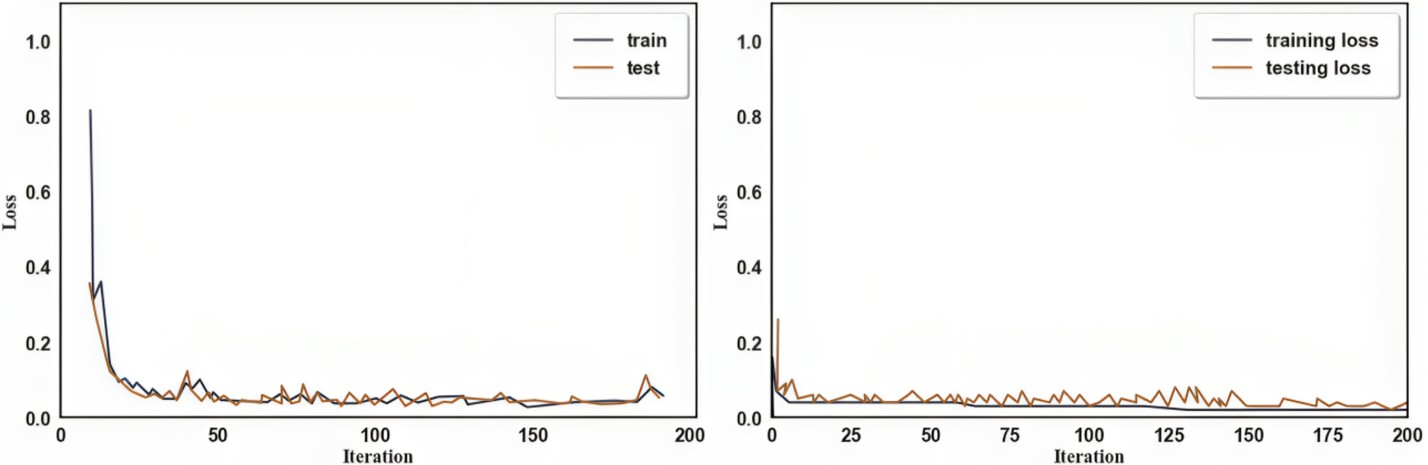

**Figure 6 Training *vs* Testing accuracy (A) training *vs* testing accuracy of PPMI dataset (B) training *vs* testing accuracy of NTUA dataset.**

chance that the network might be either under or over-fitted. We ran the network through 200 epochs of training on these datasets. The suggested model's training and testing accuracy spanned between 0.98 and 0.99. The hyperparameter configuration is shown in Table 1.

In Figs. 6 and 7, the graphs show the categorization accuracy and loss value of the IDS concerning the number of iterations. As observed in the image, the approach used in this study had a positive convergence effect. We divided the dataset into two halves for the model's training and testing. A total of 200 training epochs of the processed training set were used to train the suggested strategy during this phase. It was configured to learn at a rate of 0.01.

It employed L1 or L2 regularization techniques to penalize large weights in the network. This helps prevent the model from fitting the noise in the training data. Dropout is another regularization technique that randomly drops a fraction of neurons during training to reduce overfitting. Consider reducing the depth and width of your MobileNet model. It can use smaller variants like MobileNetV3 or even explore custom architectures tailored to the specific task.

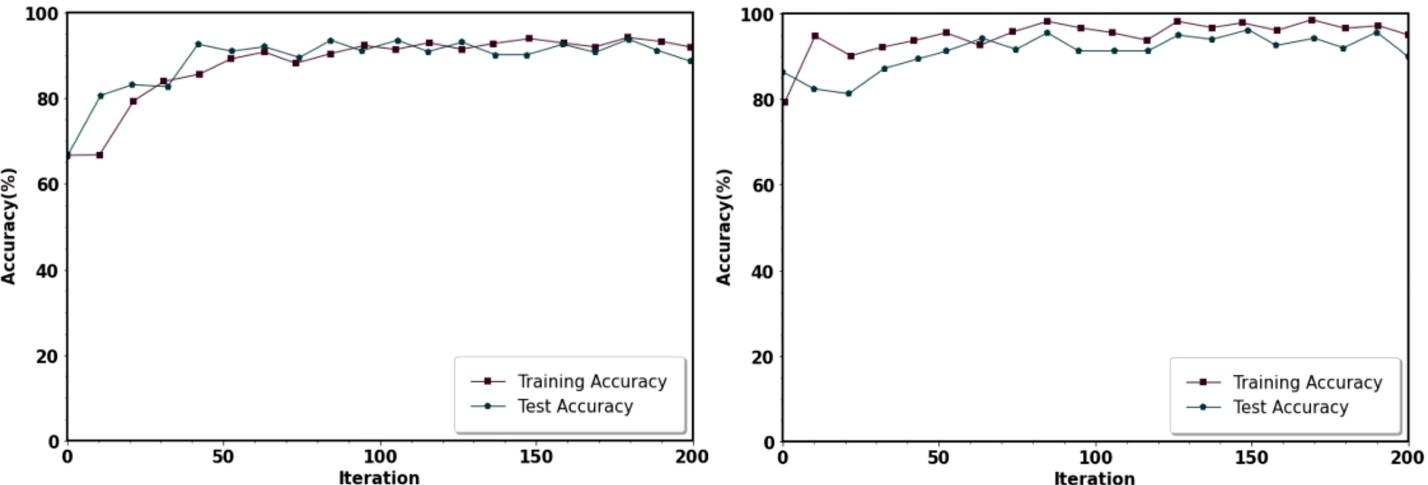

**Figure 7** Training *vs* Testing loss (A) training *vs* testing loss of PPMI dataset (B) training *vs* testing loss of NTUA dataset.

## Differentiation of existing machine learning approaches *vs* proposed

In this section, we compared our proposed deep learning approach with existing machine learning approaches. The existing machine learning approaches like SVM, RVM, and Naïve Bayes were analyzed.

We found that the suggested strategy outperformed existing approaches, demonstrating 99.06% accuracy, followed by SVM, GA-ELM, RVM, Decision tree, Naive Bayes, ANN, and CNN with 92.35%, 89.22%, 90%, 92%, 93%, 94%, and 96%, respectively. The approach proposed in this study provides a better outcome when compared to other existing solutions. CNN was the second strategy that produced a greater performance. GA-ELM performed less effectively than the other strategies. A comparison of existing approaches is shown in Fig. 8 and Table 2.

## Comparison of transfer learning approaches

The existing transfer learning approaches like DenseNet121, VGG16, ResNet, MobileNet, and Inception V3 were differentiated with the proposed method. The table compares existing transfer learning approaches with the proposed ones.

F-score, sensitivity, accuracy, and specificity were also used to compare the results. In Table 3, the comparison findings for the transfer learning approach are presented. Compared with the other approaches, the proposed method yielded a superior performance. Figure 9 shows the comparison of existing approaches with the proposed method.

The receiver operating characteristic (ROC) curve shows the performance of a model used for categorization across all levels. Two variables, TPR and FPR, were plotted on this curve. The ROC curve is shown in Fig. 10.
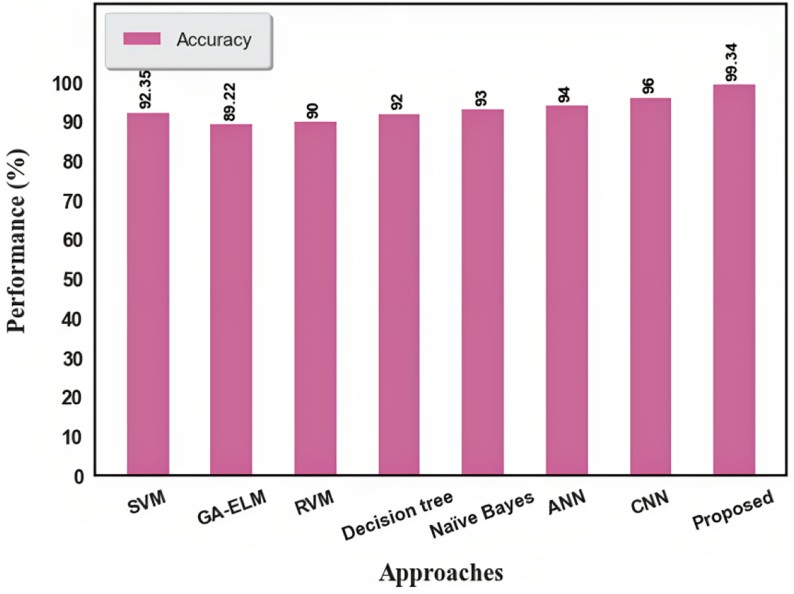

Figure 8 Comparison of existing approaches.

**Table 2 Comparison of existing approaches.**

| Approach | Dataset type | Accuracy |
|---|---|---|
| SVM | MRI | 92.35% |
| GA-ELM | MRI | 89.22% |
| RVM | PET scans | 90% |
| CNN | MRI scans | 96% |
| Decision tree | 3-T MR imaging | 92% |
| Naïve bayes | MRI | 93% |
| ANN | SPECT and TRODAT imaging | 94% |
| Proposed | 3-T weighting images and MRI scans | 99.06% |

**Table 3 Comparison of existing approaches with proposed approach.**

| Model | Accuracy (%) | Sensitivity (%) | Specificity (%) | F1-score (%) | AUC (%) |
|---|---|---|---|---|---|
| DenseNet 121 | 93.31 | 90.02 | 86.05 | 87.99 | 90.71 |
| VGG 16 | 97.65 | 88.06 | 79.30 | 83.45 | 89.36 |
| ResNet | 97.13 | 76.09 | 88.17 | 81.68 | 93.76 |
| MobileNet | 96.35 | 90.02 | 86 | 87.96 | 98.74 |
| Inception V3 | 98.49 | 84.04 | 81.78 | 82.89 | 97.13 |
| MobileNet V3 | 99.34 | 98.53 | 99.12 | 98.82 | 99.06 |

Table 4 shows the comparison of features. The essential features were extracted based on the PCFA approach. Our proposed approach extracted all the features. Compared with the previous methods, PCFA performed well.

**Peer**J Computer Science

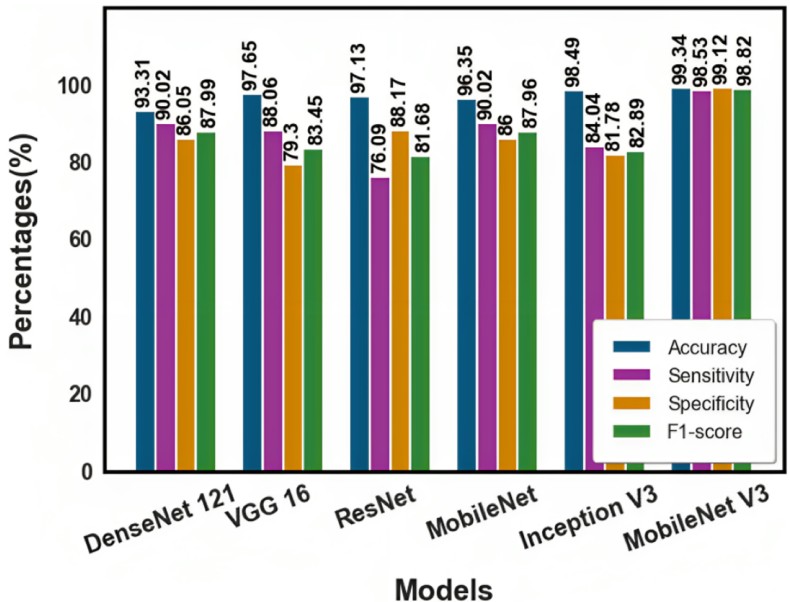

**Figure 9 Comparison of existing approaches with proposed method.**

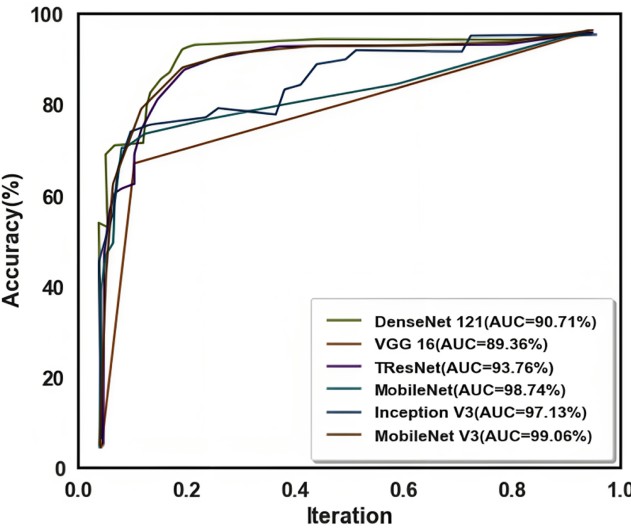

**Figure 10 Receiver operating characteristic (ROC) curve.**

Comparison of existing machine learning approach with proposed deep learning approach is shown in Table 5. The existing machine learning approaches like neural network, naïve Bayes, random forest, and SVM were employed to compare. While comparing with all four methods, the proposed approach exhibited greater performances over accuracy, sensitivity, and specificity.

**Table 4 Comparison of extracting features.**

| Features | AUC | Accuracy | Sensitivity | Specificity |
|---|---|---|---|---|
| NF | 0.94 ± 0.01 | 0.88 ± 0.06 | 0.85 ± 0.09 | 0.88 ± 0.09 |
| VBM+CF | 0.93 ± 0.04 | 0.86 ± 0.06 | 0.88 ± 0.08 | 086 ± 0.08 |
| NF+CF features | 0.97 ± 0.02 | 0.93 ± 0.04 | 0.93 ± 0.06 | 0.92 ± 0.07 |
| FS+CF | 0.82 ± 0.06 | 0.72 ± 0.07 | 0.74 ± 0.10 | 0.71 ± 0.12 |
| FS | 0.70 ± 0.06 | 0.63 ± 0.07 | 0.60 ± 0.11 | 0.66 ± 0.11 |
| VBM | 0.87 ± 0.05 | 0.79 ± 0.08 | 0.77 ± 0.12 | 0.77 ± 0.11 |

**Table 5 Proposed deep learning approach with existing machine learning.**

| Method | AUC | Accuracy | Sensitivity | Specificity |
|---|---|---|---|---|
| Neural network | 0.94 ± 0.04 | 0.89 ± 0.05 | 0.90 ± 0.08 | 0.88 ± 0.07 |
| Naïve bayes | 0.97 ± 0.03 | 0.92 ± 0.05 | 0.91 ± 0.07 | 0.93 ± 0.07 |
| Random forest | 0.97 ± 0.02 | 0.91 ± 0.05 | 0.90 ± 0.07 | 0.91 ± 0.07 |
| SVM | 0.97 ± 0.02 | 0.93 ± 0.04 | 0.93 ± 0.06 | 0.92 ± 0.07 |
| Proposed | 99.06 ± 0.07 | 99.34 ± 0.02 | 98.53 ± 0.09 | 97.68 ± 0.08 |

**Table 6 Result of optimized MobileNet V3 and MobileNet V3 during the testing process.**

| Dataset | Optimized MobileNet V3 | | MobileNet V3 | |
|---|---|---|---|---|
| | Accuracy (%) | Running time (s) | Accuracy (%) | Running time (s) |
| PPMI dataset | 99.13 | 0.24 | 98.09 | 0.46 |
| NTUA dataset | 98.36 | 0.33 | 96.53 | 0.67 |

The results of optimized MobileNet V3 and MobileNet V3 during the testing process are shown in Table 6 and Figs. 11 and 12. Before employing the optimization approach, the MobileNet V3 yielded 98.09% accuracy in the PPMI dataset and 96.53% in the NTUA dataset; the proposed approach performed slightly lower. While we used the IDMO algorithm to optimize the proposed approach, the performance improved considerably.

Training, testing accuracy, and loss comparison are shown in Table 7. The existing approaches like DenseNet, VGG 19, ResNet, and inception V3 obtained low performances compared with the proposed model. The approach proposed in this study, optimized by IDMO, showed high accuracy and a lesser loss.

A comparison of existing related works is represented in Table 8. The existing authors used various datasets, including PD, PPMI, PD audio dataset, *etc.*, but our proposed approach employed PPMI and NTUA datasets. While compared with existing approaches, the proposed approach yields a greater predictive performance. The second most performance was presented in reference (*Loh et al., 2021*).

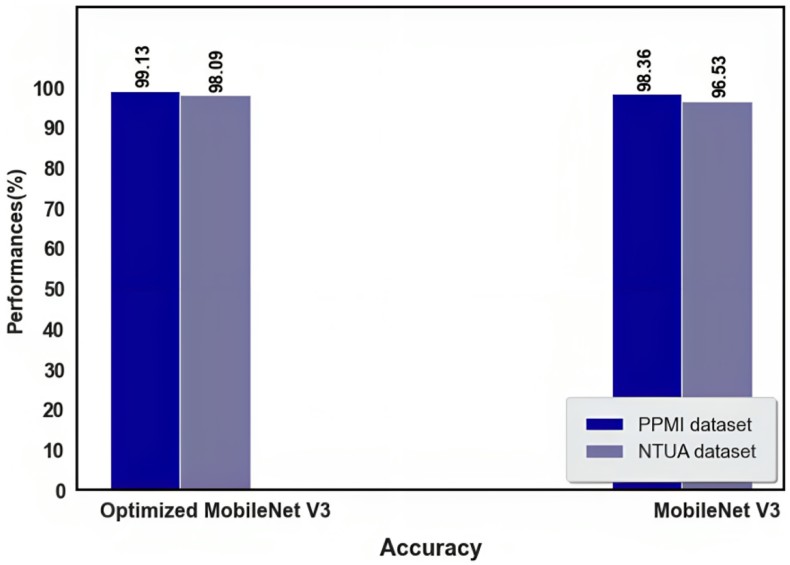

**Figure 11 Accuracy of optimized MobileNet V3 and MobileNet V3.**

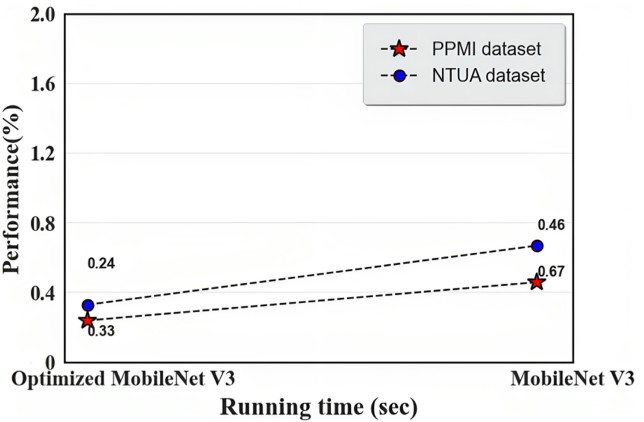

**Figure 12 Running time of optimized MobileNet V3 and MobileNet V3.**

**Table 7 Training and testing accuracy comparison with existing approaches.**

| Techniques | Training accuracy | Training loss | Testing accuracy | Testing loss |
|---|---|---|---|---|
| DenseNet | 93 | 0.061 | 87 | 0.10 |
| VGG19 | 98.64 | 0.095 | 81.25 | 0.07 |
| ResNet | 96.27 | 0.029 | 93.54 | 0.07 |
| Inception V3 | 95 | 0.038 | 84.3 | 0.112 |
| Optimized MobileNet V3 | 99.21 | 0.018 | 98.79 | 0.06 |

**Table 8 Comparison of existing research with proposed.**

| References | Method | Dataset used | Accuracy | Sensitivity | Specificity |
|---|---|---|---|---|---|
| *El Maachi, Bilodeau & Bouachir (2020)* | 1D-Convnet | PD dataset | 94.1% | 93.7% | 96.8% |
| *Loh et al. (2021)* | 2D-CNN | PD dataset | 99.06% | 98.22% | – |
| *Ortiz et al. (2019)* | Lenet 5 and AlexNet | PPMI | 95.1% | – | – |
| *Karaman et al. (2021)* | Deep CNN | mPower voice database | 89.75% | 91.50% | 88.40% |
| *Diaz et al. (2021)* | 1D CNN and BiGRU | PaHaW and NewHandPD dataset | 94.44% | 98% | 90% |
| *Olivares et al. (2020)* | ELM and BAT algorithm | Parkinson's disease audio dataset | 96.74% | – | – |
| *Vyas et al. (2022)* | 2D and 3D CNN | PPMI | 88.9% | 92% | 92% |
| Proposed | Optimized MobileNetV3 | PPMI and NTUA | 99.34% | 98.53% | 99.12% |

# CONCLUSION

Parkinson's diagnosis is an extremely challenging medical problem. Although it is technically challenging to confirm a Parkinson's diagnosis, practitioners can recognize the disorder by examining patients and looking at various symptoms. The optimized MobileNet V3 was used in the proposed study to examine MRI data to identify Parkinson's disease classes. MobileNet V3 method was optimized with the help of the Improved Dwarf Mongoose Optimization algorithm (IDMO). For extracting the features from the preprocessed image, we presented a new Pyramid channel-based feature attention network (PCFAN) that employs a multi-stage design with attention blocks at every step. Here, we used PPMA and NTUA datasets for our experimental work, then compared our proposed approach with other existing "state-of-the-art" approaches to analyze the efficiency of our work. The suggested approach performed better than the existing systems in terms of accuracy, specificity, F-score, and sensitivity, with 99.34% accuracy, 99.12%, 97.78%, and 98.53%, respectively. In previous studies, they achieved lower accuracy with higher computational time. In this research, the proposed methodology achieved higher classification accuracy than previous studies with less computational time using deep learning based feature extraction and optimized classification techniques.

Future studies will tackle three important subjects. The suggested expert system's performance will initially be evaluated using a variety of datasets. Hybridization of algorithms or new nature-inspired algorithms for feature selection can be investigated to identify PD and other applications. Also, it will be crucial to improve the variety of deep learning comparison techniques.

## Funding
The authors declare that they have no competing interests.

## Competing Interests
The authors declare that they have no competing interests.

## Author Contributions

- Sukanya Pechetti performed the experiments, performed the computation work, authored or reviewed drafts of the article, and approved the final draft.
- Battula Srinivasa Rao conceived and designed the experiments, analyzed the data, prepared figures and/or tables, and approved the final draft.

## Data Availability

The data is available at figshare: Sukanya, P.; B. Srinivasa Rao, Dr. (2023). A deep learning-based Parkinson's disease classification using fused images. figshare. Dataset. https://doi.org/10.6084/m9.figshare.23514447.v3.

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
