# Peer review of "Optimized MobileNetV3: a deep learning-based Parkinson’s disease classification using fused images"

_PeerJ Computer Science, doi:10.7717/peerj-cs.1702_

## Round 0.1 · original submission · Major Revisions

The reviewers have substantial concerns about this manuscript. The authors should provide point-to-point responses to address all the concerns and provide a revised manuscript with the revised parts being marked in different color.

**Language Note:** The review process has identified that the English language must be improved. PeerJ can provide language editing services - please contact us at [email protected] for pricing (be sure to provide your manuscript number and title). Alternatively, you should make your own arrangements to improve the language quality and provide details in your response letter. – PeerJ Staff

Reviewer 1 ·

Basic reporting

in sufficient field background provided.

Experimental design

methods described with insufficient detail l& information to replicate.

Validity of the findings

too far-fetched, not convincing.

Reviewer 2 ·

Basic reporting

The manuscript “Optimized MobileNetV3: A deep learning-based Parkinson's disease classification using fused images” proposed a model that can help with Parkinson's disease early diagnosis. Some major and minor revisions should be made by authors.

Major revisions:
1.The proposed approach obtains 99.34% accuracy, 98.53% sensitivity, 97.78% specificity, and a 99.12% F-Score compared to previous methods. This result is impressive. I am wondering if these almost-perfect results are caused by overfitting or data leakage in the model construction. I noticed that two datasets (NATU and PPMI) were used in this study. Thus, I would like to recommend a method to ensure the robustness of prediction results. For example, the author could randomly divide the NATU dataset into training (50%) and evaluation (50%) dataset. The model was trained on the training dataset and evaluated in the evaluation dataset. Then, the dataset PPMI will be used as the independent dataset to validate the performance of the model. The author should not try to normalize the PPMI with NATU datasets, which will cause data leakage. Shortly, the independent dataset will make the prediction result more solid.
2.In Part “Related Works”, the author introduced some deep learning models for PD categorization. Are there any other published models that also predict Parkinson's disease early diagnosis by NATU and PPMI? If there are models, please compare them with these models.
3.In Part “Code Availability”, the author said that it is Not applicable. This is unusual for a study of deep learning model construction. The author should upload the code used in this study to Github.


Minor revisions:
1.MobileNetv3 is a model proposed by a team of Google. The author should make sure that in the training process of MobileNetv3, the Google team did not use resources from NATU and PPMI.
2.Some figures, such as Figure 1 and Figure 2, have a strange ratio of height and width. Make sure your figures are properly presented.
3.Some figures, such as Figures 8-11, only contain one single subplot. I will recommend merging these figures into one.
4.In NATU, the author mentioned that “We utilized 29400 (70%) images from the dataset for training and the remaining 12600 (30%) for testing purposes”. How about another one (PPMI)?
5.Please carefully check the equations in this study. The equation for the calculation of Evaluation criteria, such as Accuracy seems to be wrong.

Experimental design

no comment

Validity of the findings

no comment

Reviewer 3 ·

Basic reporting

In this manuscript, the authors applied MobileNetV3-based classification to enhance the prediction performance of deep-learning-based PD diagnosis model. Comparisons were conducted to prove the proposed approaches have better prediction performance over the reported methods. The study is promising with novelty and fits the scope of PeerJ. However, some of the technical details were not thoroughly discussed, and the calculation of F1 scores is questionable. Moreover, there are also some typos and format issues in the figures and tables, the authors should prepare them with more care and review the paper before submission.
There are no page numbers in this manuscript, the following page numbers mentioned will be the pages in Adobe PDF Reader.
Page 10, Line 143-149. The author briefly mentioned the limitations of reported deep learning based PD studies. How were those limitations overcome by the authors’ approaches? It is recommended to have more discussions about it. Also, how about the prediction performance of the reported methods?
Page 10, Line 166-167. How were the processing results from two created copies used?
Page 11, Line 185. “Pi It is calculated from equation 2.” There is a typo here, the “it” should be removed.
Page 11, Line 191. What are the rationales here to set a and b as constant 0.3293 and 1.158?
Page 13, Line 227-228. “The feature maps’ resolution is reduced in half, and their depth is increased to 64 and 128….” It is not clear if the depth is increased from 64 to 128 or the resolution is reduced to 64 and the depth increased to 128.
Page 15, Line 275, “has the highest top-1 accuracy among the other models”, the “highest” is redundant here, also citations are needed to support this statement.
Page 20, Line 395-404, are the 6500 images from the scanner used together with the 42000 photos from NTUA Parkinson dataset for training and testing purposes?
Page 23, Conclusion section, please consider discussing any potential limitations of the study design, data acquisition (analysis tools and database), and future application.
Page 39, Figure 6, the authors may need to share their thoughts about why the prediction accuracy of both the training and test dataset of the NTUA data set is lower than the PPMI dataset, especially when the Iteration numbers are low (<50).
Page 41, Figure 7, figures a and b should be consistent in label and background.

Experimental design

no comment

Validity of the findings

Page 45. Figure 9, it seems the F1 score of some groups was not correctly calculated. For example, the Recall and Precision of Inception V3 are 84.04 and 81.78 respectively. The F1 score should be 2*(84.04*81.78)/(84.04+81.78) = 82.89 rather than 98.34 in Figure 9.
Page 52, Table 3, please list the full list of hyperparameter values for repeating purposes.
Page 55, Table 2, the significant digits of accuracy should be consistent for all the approaches.
Page 61, Table 5, why only the AUC, accuracy, sensitivity, and specificity of the proposed groups in percentage? Was the variance correct?

Additional comments

no comment

---

## Round 0.2 · accepted · Accept

Reviewers are satisfied with the revisions, and I concur to accept this manuscript.

Reviewer 1 ·

Basic reporting

no comment

Experimental design

no comment

Validity of the findings

no comment

Reviewer 2 ·

Basic reporting

\My questions have been answered by the authors, and I think this manuscript could be accepted.

Experimental design

My questions have been answered by the authors, and I think this manuscript could be accepted.

Validity of the findings

My questions have been answered by the authors, and I think this manuscript could be accepted.

Additional comments

My questions have been answered by the authors, and I think this manuscript could be accepted.